

# Technical Note: Low Cost Mesocosms Design for Studies of Tropical Marine Environments

Ruben R. Raygosa-Barahona[1],Sebastien Putzeys[1], Jorge Herrera[1] ,Daniel Pech[2]

[1]Cinvestav, Merida,97600,Mexico
[2]Ecosur,Campeche,24500, Mexico

*Correspondence to*: Ruben R. Raygosa-Barahona (r.raygosa@gmail.com)

**Abstract.** Mesocosms are an alternative to in situ ocean environmental studies which are very difficult to implement due to the challenges that the aquatic environment impose. The design of a mesocosm should consider as many variables as possible of the ecosystem to be studied, in order to obtain results that are similar to those that would be obtained in the environment.
The effects of tropical climatic conditions on a mesocosm enclosure were studied in order to evaluate their possible influence on the biological community. The mesocosm was equipped with an electric marine thruster as a means of avoiding stratification in the water  contained in it. Also, the system is submerged in water to increase the thermal inertia and maintain the temperature variations within reasonable ranges. The design does not include auxiliary forcing cooling systems. The results revealed the influence of climatology on the mesocosms' temperature and showed the feasibility of the proposed design in tropical
environments. With high variations of ambient temperature (>20ºC, during the day), the variations in the mesocosm temperature were only 3ºC.  The range of temperature variations were similar to those that occur in certain tropical environments.

## 1 Introduction

*In situ* ocean environmental studies are expensive and often difficult to perform (Reilly, 1999). On the other hand, the results
of small-scale laboratory experiments are cheap, but they do not incorporate sufficient environmental realism (Cappello and Yakimov, 2010).

The mesocosms studies are an intermediate alternative between both laboratories and *in situ* experiments. The ability to control environmental variables (Vallino, 2000) jointly with the replicability and repeatability compared to field studies (Cappello and Yakimov, 2010; Petersen and Hastings, 2001)(Auffan et al., 2018) convert the mesocosm into a powerful experimental option.
Best of all, parts (populations) and wholes (ecosystems) can be investigated simultaneously (Odum, 1984). The terms micro-, meso- and macrocosms are often used to define multifactorial ecosystems studies.  A volume classification has been suggested in which microcosms are enclosures smaller than 1 $m^3$ , mesocosms to contain between 1 to 1000 $m^3$, and finally macrocosms to contain more than 1000 $m^3$ (Culp et al., 1988). The material and design used in these cosms constructions usually depend on the experiment's localization and the objective of the study. Enclosures with flexible walls are commonly used *in situ*,



whereas containers with rigid walls are used either on shore or in indoor facilities. Mesocosms experimental approach is arguably the single most powerful method to obtain a mechanistic quantitative understanding of ecosystem-level impacts of stressors on complex systems. Most of the mesocosm facilities are located in the northern hemisphere (see MESOAQUA and AQUACOSM websites) and only a few studies were performed in tropical areas (Carneiro *et al.,* 2013), usually using flexible

enclosures (Table 1)

In the existing literature, temperature control or register was not considered  a priority despite the variation of temperature in tropical areas. As is shown in table 1, the differences in temperature recorded in the studies are highly variable, probably causing the presence of a thermocline during the experiment. None of these studies has considered temperature as a relevant factor, probably due to the small impact expected on the objective of the study.

Here we designed and tested a ground mesocosms unit system developed for studies in a tropical environment where the water column temperature profile could be a relevant factor modifying the response of the productivity and metabolic activity of the micro-organisms on the water column. Additionally the atmospheric temperature could induce marked changes in the water column temperature and high rates of evaporation.

The design of a mesocosm experiment is a complex procedure in which the variables and parameters (chemical, physical or

biological) have to be balanced with the objectives of the study and the development feasibility (Riebesell *et al.*, 2013),(Fernández-pascual et al., 2017). In the case of experiments in tropical areas such as the south of the Gulf of Mexico, the greatest challenge could be the harsh environmental conditions: the high environmental temperatures (32-40 ℃), the high evaporation rates, recurrence of storms and hurricanes (June to November) (i.e.  That make it imperative to take into account some specific considerations prior to the experimental phase. Here we describe and discuss the series of analyses and tests

performed to determine the feasibility of mesocosms experiments in the tropical region of the Yucatan peninsula, México.

Metabolism is the process by which energy and mass are transformed within an organism and exchanged between the organism and its environment. Metabolic theory predicts how metabolic rate controls ecological processes (Gillooly et al., 2001) . On the other hand,  temperature is a master variable that controls biological activity through its fundamental effect on metabolic rate  (Arrhenius, 1889);(Gillooly et al., 1994).  Temperature changes affect several physical and chemical properties

of water such as conductivity, salinity, density (viscosity), oxygen solubility, pH and carbon dioxide solubility. Also, all the biological processes are temperature dependent. The organisms typically exhibit higher respiration rates and for phytoplankton, higher primary production rates (Brown et al., 2004) mirrored by high rates of nutrient and carbon cycles.  After a 20-25 °C threshold, however, the temperature generally influences respiration more than primary production  (Yvon-durocher et al., 2015). Temperature also has the potential to shape the distribution and diversity of the marine microbial community. (Boyd et

al., 2013; Flombaum et al., 2013; Pittera et al., 2014; Thomas et al., 2012; Yung et al., 2015)  When the thermal limits are exceeded, changes in the microbial community fitness are produced, with cascading impacts on the ecosystem services (Costanza et al., 1997, 2014). Finally, the temperature also affects the toxicity of compounds such as heavy metals or ammonium ,(Cairns, 1975). Metabolic rates increase with temperature exponentially according to the Boltzmann-Arrhenius Function:



$$B = ae^{-Ei/kT} \qquad (1)$$

Where B is the mass-specific metabolic rate (in units of time$^{-1)}$ (Marañón et al., 2018), k is the Boltzmann constant ($8.62 \times 10^{-5}$ eV K$^{-1}$), T is the Temperature in K, a is a normalization constant and Ei is the activation energy (eV). Ei of basal metabolic rate (maintenance respiration) maintains relatively similar values (0.6–0.7 eV) across all organisms from microbes to plants and animals (Brown et al., 2004)(López-Sandoval et al., 2010). For each taxon, *B/a=Bo* is approximately independent of *a;* then almost all the effects on temperature variations are contained in the normalization term *Bo* (Marañón et al., 2018).

$$B_o = e^{-Ei/kT} \qquad (2)$$

The value of *B* at some temperature *T* could be related to its value at some other temperature *To* by

$$\frac{B(T)}{B(To)} = \frac{Bo(T)}{Bo(To)} = e^{Ei(T-To)/kTTo} \qquad (3)$$

Equation (3) states that the changes in metabolic rates *B,* caused by a temperature variation depends on both: the amount of temperature variation (*T-To*) and on the initial temperature *To*. Figure (1) shows the variation of the normalized metabolic rate , (*B(T)/B(To)*) with respect to the temperature starting from several initial temperatures (*To*). For lower *To,* similar increments of temperature produce greater variations in the growth rate than higher *To* temperatures. For a given *To*=20 ° C, a temperature increase of 3 °C will produce a variation in the growth rate of 30% while for a *To=25* ° C a similar increment of temperature will produce a variation in the growth rate of 25%. This example shows that in a mesocosms it is very important to keep temperature variations controlled in at least the temperature ranges that occur in the environmental conditions of the site where the samples to be studied are to be taken. Probably the temperature control is the primary challenge with on-land mesocosm experiments due to the thermal inertia of the aquatic environment (Leblud et al., 2014). For example, to maintain a constant temperature of 15°C in mesocosms of 2500 liters of saline water (~36 PSU), it is necessary to apply an energy of approximately 10 $^{-6}$ J for each °C above the environmental temperature. The energy could be applied using a resistive heater controller that heats the water column when necessary. In the opposite case, when the water column needs to be cooled, as in our case the situation turns on several difficulties.

The efficiency of any refrigerant system depends, among other things, on the coolant used, the thermal insulation between the medium and the surrounding environment and of course the refrigerant system used, (Fleming, 2015). Besides, the design and selection of any refrigeration system is a tradeoff between the system cost, energy cost and facilities capacity available, the volume of the substance to be cooled and the power applied to bring the water to the desired temperature below the ambient temperature during the time of the study.



Whichever the case is, heating or cooling, the effect of the temperature on the entire volume is not instantaneous. During the heating or cooling process, a thermocline can appear, causing an alteration and forming micro-conditions that can alter the response of the biological communities during the experiment. The way to prevent stratification and to ensure a more efficient heat transfer to the entire water column is by using a circulation system. Traditionally pumps (centrifugal or peripheral) used

in circulation systems need to accelerate the liquid at relatively high speeds using high differential pressure levels. In mesocosms systems the high velocities could promote a stress on the communities or modification due to the destruction of the more delicate cells, affecting the experimental conditions. The alternative is to use a marine thruster device individually located in each mesocosms unit. The thruster propeller can rotate at speeds much lower than the impellers of the water pumps. For example, a water pump rotates typically at 1800 rpm while a marine thruster works at speeds of 700 rpm at maximum

velocity.

Whatever the method used to promote the water motion inside the cosms it is necessary to know the profile of water velocities in order is assess the potential stress or damage to marine communities under study. Although instruments exist to measure velocity profiles in the aquatic environment such as Aquadopp profiler from Nortek, or ADP from Sontek,  their capacity exceeds what is needed in a mesocosm (Nortek, 2010),(Sontek, 2015). As an alternative, a computer simulation using

Computational Fluid Dynamics software (CFD) can be employed. CFD software is commonly used in mechatronic design which involves marine thrusters (Maciel, P., Koop, A. and Vaz, 2013).

Here we show and discuss the importance of including a profile of velocities measurements on a ground mesocosms using a CFD to make spot measurements of speed prior to experiments. It is also proposed to measure the temperature along the water column of the mesocosm subject to various internal and external  conditions to verify how they affect the variations throughout

the day of the environmental conditions within the mesocosms.

## 2 Design and Performance Evaluation.

### 2.1 Mesocosm Design.

Due to the importance of temperature in altering the rates of metabolism, it is necessary that the temperature in the mesocosms be maintained within the range in which it would be in the ecosystem to be studied. In order to determine the appropriate

temperature range, the results of sampling in two places are presented below.

The first site of study is Telchac  Beach  (21°20'25.02" N  89°18'25.56" W), marked with a black dot on the map, figure 2. The data collection was performed with a YSI probe .  Figure 3 presents a record of the temperatures over 2 days during the month of May 2016 of seawater in Telchac Harbor in the Gulf of Mexico. The probe was installed at  a depth of 1.5 m while the ambient temperature was recorded with a Vaisala wxt520 Meteorological Station. In these graphs it can be seen that

water temperature variations occurred in synchrony with variations in air temperature. The variations are of the order of 4 °C. Therefore it is reasonable to consider that a temperature variation in a mesocosm for these tropical conditions should not exceed these limits.



By contrast in figure 5  the temperature record taken at a distance of 10 kms to the north  at 9 m  depth over a week shows that the variations in seawater temperature are not necessarily in sync with the ambient temperature and fluctuations over a day are not greater than 1 °C.

In the previous examples, the sampling points are located less than 10 km away from each other. Important variations in the

behavior of the temperature are shown, despite its proximity. However other conditions such as depth, currents and seasonal variations can produce significant changes in sea temperature. The variation in magnitude of these variables must be taken into account to verify that the mesocosm presents conditions similar to those of the place of study.

In tropical regions such as the south of Gulf of Mexico (GoM) the variations of ambient temperature could be higher. This is the main characteristic for using an auxiliary heating or cooling systems(France and Duffy, 2006)(France & Duffy,

2006)(France & Duffy, 2006)(France & Duffy, 2006)to increase the thermal capacity of the system (France & Duffy, 2006). The containers used for the ground mesocosms prototype are commercial water tanks made of high-density polyethylene. The container shape is a cylinder  1.75 m high by 1.55 m in diameter, see Figure 5.  A commercial brushed NEREAUS marine electric thruster of  ½ hp composes the recirculating system.  The thruster's velocity is driven by a PWM motor controller either controlled by a microcontroller or manually by a potentiometer adjustable by the user.

Note in Figure 5 the temperature column sensor located on the left side. Additionally, the system has a radio link to be able to monitor the data in real time.

**2.2 Performance tests.**

To evaluate the behavior of the water column in the  mesocosm, it was subjected to a series of tests that included the temperature measurement in the water column subjecting the system to different conditions.

**2.2.1 Homogenization test.**

To evaluate the ability to avoid stratification by the use of an electric thruster, was created an artificial temperature gradient using two heaters of 2000 W each. The heaters were placed 30 cm below the surface for an hour, and 15 temperature sensors were installed vertically aligned along the water column, one of the sensors was installed just above the water level and one more 30 cm above to measure the ambient temperature. The electric marine thruster was then power on at 25% of its maximum

power.  Figure 6 a)  shows the temporal evolution of the temperature in the container throughout the experiment. From now on for the following figures, S1 to S15 labels and their respective color stand for each one of the signal fetched for  15 temperature sensor installed into the container. The sensors 1 to 13 are submerged in the water in an order according to their number from the deepest to the surface. The sensors 14 and 15, in blue and brow respectively are above the water level. At time t = 12:24:30 the electric thruster is turned on. It was observed that the system takes approximately 1 minute to homogenize,

even though the external temperature continues to increase.

**2.2.2 Immersing the mesocosms into water  to increase their thermal capacity.**

To evaluate the effect of inmersing the mesocosms the following tests were performed:

c) the mesocosms were exposed to the environment to observe the temperature variations as well as the stratification.

d) the mesocosms were immersed in water.



e) the effects of the thruster were included, while immersing in water the mesocosms.

e) the thruster was used with different on / off cycles in.

As a first step  the system was left to evolve following the climatic conditions,  Figure 6 b) shows this evolution during the day-night temperature cycle and the development of a stratification throughout the day period. The registered temperature on all sensors converges until a minimum temperature of 30ºC. Notice that the ambient temperature goes from 25 °C to 45 °C. It can be noted that the stratification had a duration of 10 hours with an instantaneous variation of maximum temperature of 3 °C and a maximum change interval of 4 °C.

The test was repeated with the mesocosm inmersed into water.  Figure 6 c) shows as expected, the formation of a stratification. However, the variation of temperature was reduced.  It can be shown that even though the variations in environmental temperature were similar to those in the previous experiment, the temperature variation in the water column is lower with a maximum of 2.5ºC.

### 2.2.3 Including the Electric Marine Thruster

Once the presence of water temperature stratification was detected, the next step was to include an electric marine thruster to eliminate the presence of water stratification. The continuous functioning of the marine thruster at 2 m s$^{-1}$   was sufficient to eliminate the water stratification effect (Fig 6 d). No water stratification occurred and the change in temperature over a night-day cycle was 4 ºC contrasting with the 25 ºC of environmental temperature variation.

A disadvantage of brushed motors such as those used in certain electric thrusters  is the wear of some of their internal components related to the use. This feature suggests interleaving on/off cycles. Figure 6 e) shows the temperature behavior when using  propellers with 5 minute on/off cycle. It can be noted that superficial layer started to heat during this short time, compared to the deeper layer. Figure 6 f ) shows the experiment with the motor at 20 % of its maximum power with a duty cycle of 10 minutes. There are small undulations along the lines of temperature produced by the effect of the motor. Comparing with the previous case, it is concluded that a 50% duty cycle with a period of 5 minutes of ignition for 5 minutes of shutdown almost produces the same effect as using the motor continuously. In the case of prolonged periods of test, this procedure can significantly extend the life of the motor brushes. On the other hand, the use of a work cycle of 10 minutes presented stratification and therefore its use is not recommended.

### 2.3 Water Velocities Profiles Produced by the Thruster.

The use of marine thrusters has the additional advantage of avoiding the accumulation of sediments and maintaining the homogeneous conditions in the entire volume of water. The velocity of the electrical thruster can be easily controlled and could be varied over a wide range of speeds including low, by controlling the current that feeds them (Bishop, 2006).  A computer numerical simulation using the CFD software Solidworks © shows the velocities profile and flow of a virtual liquid (water) at speeds of rotation of a propeller of less than 100 rpm (Figure 7). Results from the simulation show regions with maximum velocities in the center of the mesocosms (0.5 m s$^{-1}$ ) while lower speeds occur at the edge ( 0.15 m s$^{-1}$ ) . The mathematical model of a thruster can consist of more than 40 parameters and is extremely nonlinear (Bachmayer et al., 2000). That is, a



small variation in some parameter could produce a very different result. Therefore we verify the velocities using a spot measure and compare them with those obtained by the CFD.

### 2.3.1 Water Velocity against power supplied.

To measure the speed, a propeller based flow meter was placed at approximately 15 centimeters from the center of the thruster
propeller and subsequently moved to the edge of the mesocosms. In Figure 8 maximum power (100%) corresponds to ½ hp, while the velocity was in m s$^{-1}$. Vel1 stand for the velocity in the center of the mesocosms while vel2 stand for the velocity at the edge. The power was incremented in steps of 10% of the maximum power. It can be observed that at a power greater than 50% there is no significant increase in the water speed, although more turbulence was observed. On the other hand the values obtained with the CFD in the center and the edge of the mesocosm, 0.5 and 0.15 m s$^{-1}$ correspond to the values obtained
by direct measurement of 0.5 and 0.125 m s$^{-1}$ respectively.

### 3 Discussions and Conclusions

No literature describing the use of electric propellers in mesocosms was found for tropical onshore mesocosms. Without the use of a thruster, a significant temperature difference appears between superficial and bottom layers during a 24h period. This temperature difference could promote micro-conditions. (Nada Krstulovic, at al 1995) that would alter biological
communities' responses during an experiment, mainly when performed on bacterioplankton and phytoplankton (Lorena Grubisic at al 2012)

The use of electric thruster systems avoided stratification efficiently on a tropical offshore mesocosms system. Additionally, its inclusion in the mesocosms design can be useful to maintain the homogenization of the water column helping to reduce the stress of marine communities under study compared to using water recirculation systems. The use of marine motors also
presents the additional advantage of avoiding the accumulation of organic matter in the bottom of the tanks and thus avoiding the development of low oxygen conditions.

The different tests carried out compared to the *in situ* data suggest the variability inside the mesocosms tank could be considered as part of the natural daily changes.

Also, the use of CFD could be used to estimate the velocity profiles inside a mesocosm to evaluate the stress which the use of
the thruster could produce in the communities under study.

The different tests have been carried out under no controlled environmental conditions, Surprisingly, environmental variations of up to 25 °C of temperature resulted in only 3 °C of temperature variation in the mesocosms. The authors consider that this variability compared to the *in situ* data suggest the variability inside the tank could be considered as part of the natural daily changes in the environment.
The immersion of the mesocosm unit in the water together with the use of a marine thruster shows its efficiency to maintain water temperature relatively stable without the need to add auxiliary refrigeration equipment. This is a lower cost strategy. During the preparation of the different tests, a decrease in water level due to evaporation was observed, averaging 6 mm per day. This reduction in level represents approximately 6 liters of water per day. This variation can produce important changes in some physical parameters such as salinity. Future work may be to implement a system for measuring evaporation.



The series of tests performed allow us to conclude that in the case of onshore mesocosms experiments in tropical areas, auxiliary and higly costly refrigeration equipment is not necessary. This allows both lower installation and running costs.

**Acknowledgements**

Thanks to our colleagues from the ISMER-UQAR, specially to Gustavo Ferreyra for share it experiences with the implementation of on land mesocosms. And to Ismael Mariñp Tapia for share records from sea temperatures in Telchac Beach. This work was part of the binational collaboration México-Quebec project "Desarrollo de experimentos en mesocosmos para evaluar la vulnerabilidad de los ecosistema marios ocasionada por la actividad petrolera: comparición latitudinal" FONCICyT 265435. Financial support was also provided by the SENER2012-1-Hidrocarburos (Sectorial Research Funds 0020SRE-CONACYT S0018) from the Mexican Council of Science and Technology (ConaCyt), the Mexican Secretary of Energy (SENER) and the Mexican Petroleum Company (PEMEX).

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

| Water type | Objective(s) under study | Temperature regulation | Temperature sampling | Sampling frequency | Temperature variation | Reference |
|---|---|---|---|---|---|---|
| freshwater | Fish-phytoplankton | None – *in situ* | No | - | No referenced | Do Rêgo Monteiro Starling, 1993 |
| freshwater | Predator - prey | None – *in situ* | yes | 3 times during the 16 days | 25.6-29.8ºC | Maioli Castilho-Noll and Arcifa, 2007 |
| freshwater | Species loss influence on community structure and ecosystem multifunctionality | None – *in situ* | No | - | No referenced | Pendleton *et al.*, 2014 |
| freshwater | Foodweb configuration and nutrient sources influence on the limnological variables | None – *in situ* | No | - | No referenced | Carneiro *et al.*, 2013 |
| brackish | Predator – prey | None – *in situ* | yes | 2 times per day | 22-28ºC | Humphries *et al.*, 2011 |
| brackish | Hydrocarbon impact on bacteria and phytoplankton | None – *in situ* | Yes | - | 30.2+-0.4ºC | Nayar *et al.*, 2005 |

**Table 1. Experimental conditions and temperature environmental control in literature of tropical mesocosms.**


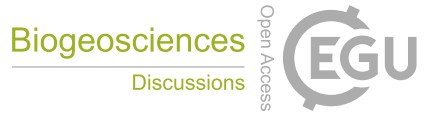

**Figure 1.** **Variation of the normalized metabolic rate,** $(B(T)/B(To))$ **with respect to the temperature starting from several initial temperatures** $(To)$**.**

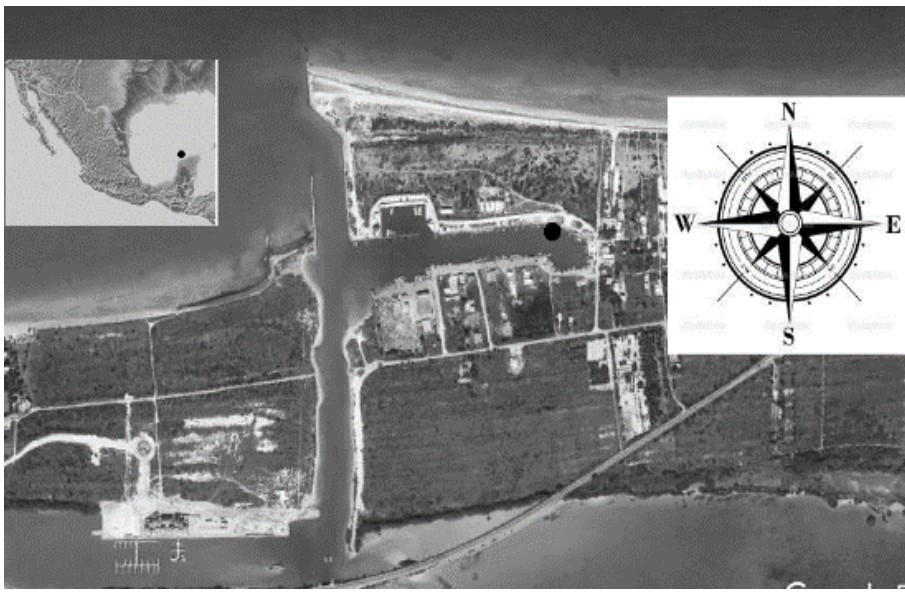

**Figure 2 Geographical localization of Telchac Harbor (21°20'25.02'' N 89°18'25.56'' W).**



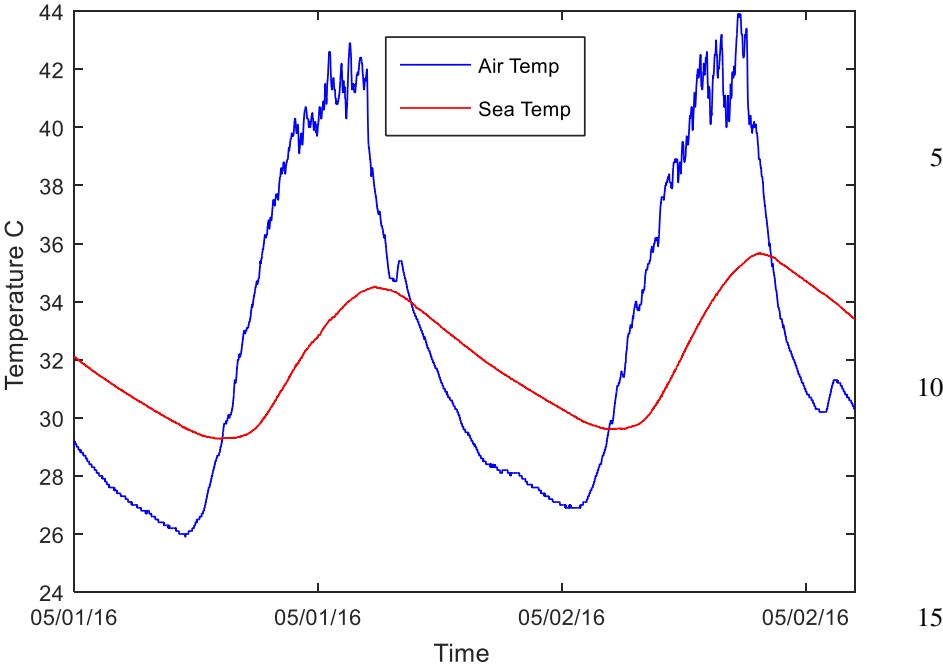

**Figure 3    Variations of sea and air temperatures in  Telchac Harbor during two typical days in May 2018.**

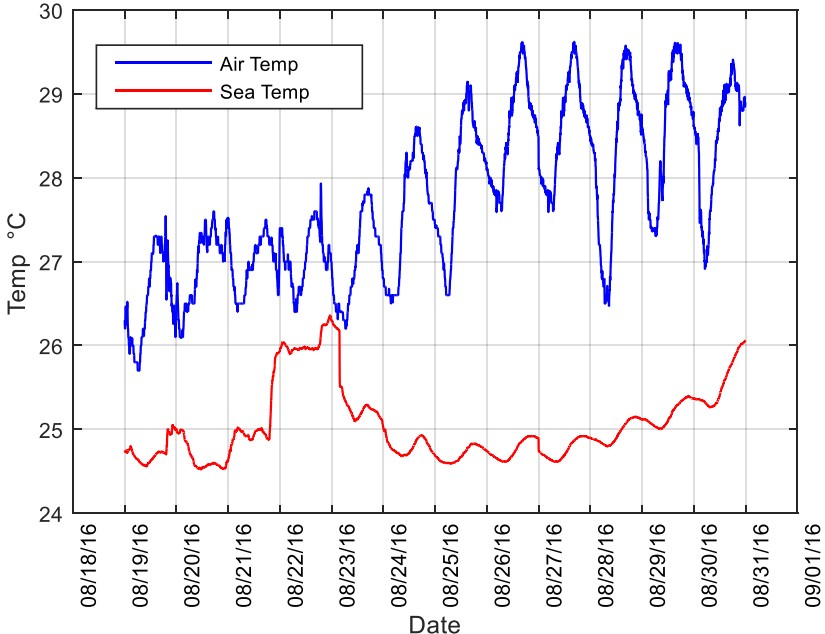

**Fig 4.  Sea temperature variations registered at 9 m depth during August 2016.**



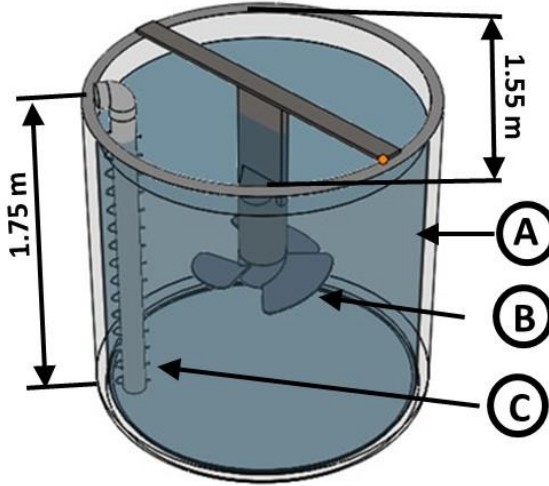

**Fig. 5  mesocosm schematic design: A) container, B) propeller, C) temperature sensors.**



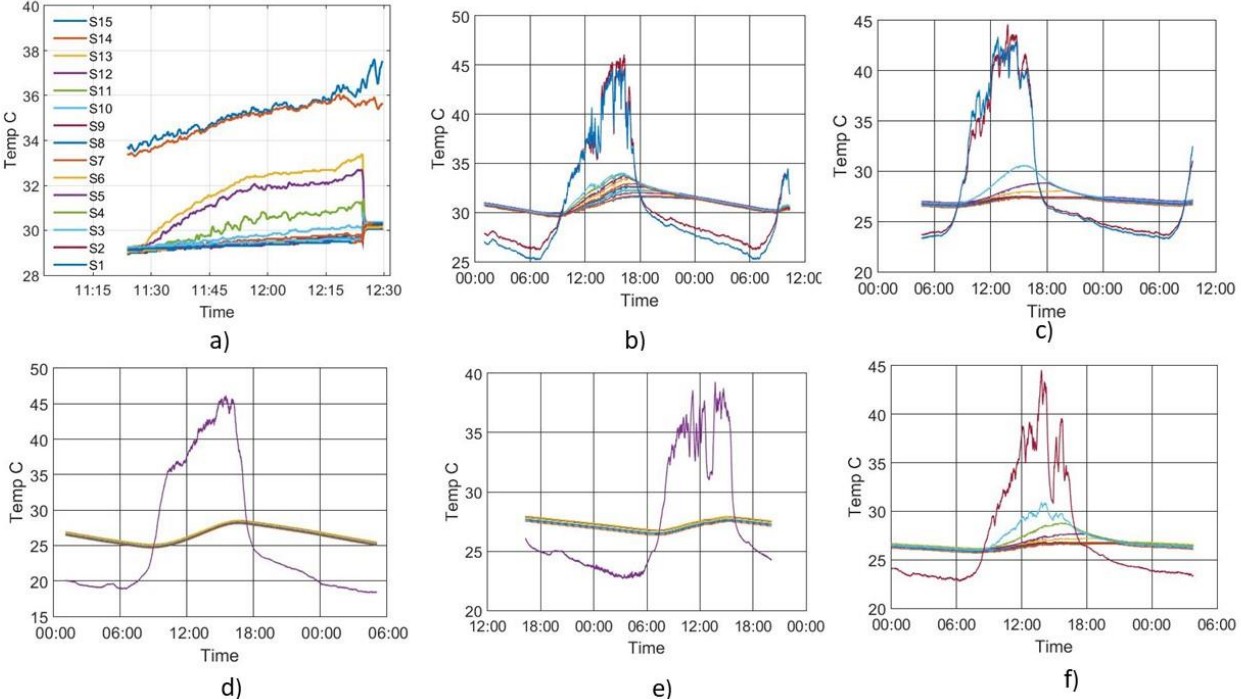

**Fig.6** **evolution of the temperature in the container throughout the experiment. a) An artificial temperature gradient was created using two heaters during along 2 hours. S1 to S15 stand for each one of 15 temperature sensor installed into the container submerged in the water in an order according to their number from the deepest to the surface. The sensors 14 and 15, in blue and brow respectively are above the water level. At 12:24:30 the thruster was turned on producing the stratification to vanish. b) the system was left to evolve following the climatic conditions .The stratification process had a duration of about 10 hours with an instantaneous variation of the maximum temperature of 3 °C and a maximum change interval of 4 °C. c) the variations in temperature when mesocosms are immersed in water. Variation in the water column on mesocosm was approximately 2.5 °C. d) effect of the thruster. e) effect of the thruster, functioning with 5 min. power cycles. f). Effect of the marine thruster, functioning with 10 min. Power cycles. There are small undulations along the lines of temperature produced by the thruster operation.**





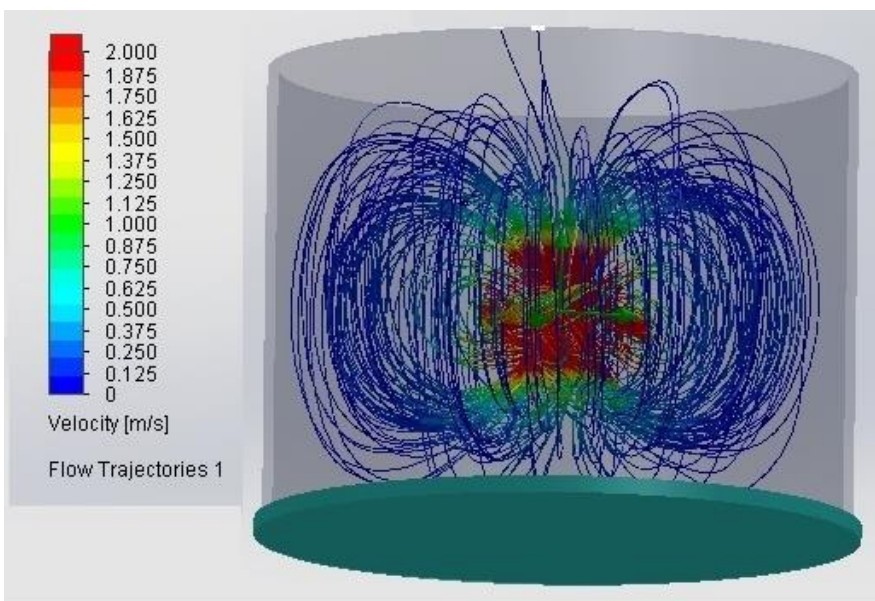

**Figure. 7 the flow of a virtual liquid obtained using SolidWorks © a CFD software.**

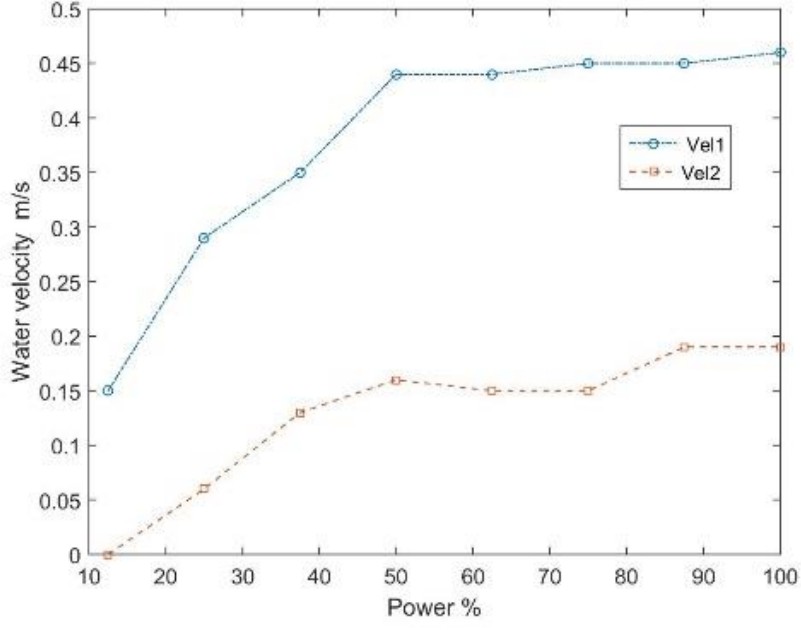

**Figure 8. Truster Power vs. water velocity.**