# Peer review of "Technical Note: Low Cost Mesocosms Design for Studies of Tropical Marine Environments"

_Biogeosciences, 2019_

## Referee Comment (RC1) · Anonymous Referee #1 · 2 Apr 2019

Teh current manuscript is a kind of technical report, not scientific article. The manuscript introduce the use of electirc thruster system to avid water stratification. Some results aout the test compared to the in situ data suggest the temperature variability could be considered within the natural daily changes. However there is any chemical and biological data in the mesocoms tank in this study. Of course seawater temperature is very important environmental factor, as author said. But I would like to suggest additional data regarding to the marine biogeochemisty such as Chl. a and phytoplankton species composition and biomass, and pH, POC and DOC etc.

---

## Author Comment (AC1) · 6 Apr 2019

The reviewer is right. Indeed, is a technical note as indicated in the title. Our main intention was to show and discuss the main considerations when designing a low-cost on-land mesocosms facilities, not the results of the experiments. As we stated in the introduction the design of ground mesocosms facilities in tropical environments is challenging due to the harsh environment condition such as high temperatures (32-40C).

Our results showed that without any of the technical improvements the mesocosms water column could reach temperatures of 45 C (in 10 hrs)(fig 6 in the draft paper) which

represents 13 C above the maximum values observed in the natural daily changes. The inclusion of the electric marine thruster and the immersion of the mesocosms unit in a water bath was useful to maintain the water temperature in a range of variability similar to the observed in the field.

Results of the experiments, including the information of the measured physical, chemical, and biological parameters will be elsewhere published.

---

## Referee Comment (RC2) · Anonymous Referee #2 · 8 May 2019

The article by Raygosa-Barahona et al. presents some useful information about the technicalities of mesocosm studies in tropical environments. As such, it is indeed well suited for a technical note. However, in its current form, the manuscript feels rushed and unpolished, containing many mistakes, inconsistencies and omissions, thus requiring refinements. If the authors were to correct the irregularities and add information on the water immersion of their mesocosms (pictures, schematics or figures), it would greatly improve the manuscript. In the context of a technical note, the introduction on the biological ramifications of temperature control may be expendable to allow the authors to focus on the technical aspects.

[Figure]

Specific comments: P1. Line 7: Mesocosms are often deployed in situ, thus are not necessarily an alternative to in situ studies.

P2.L4: Is there really no tropical mesocosm experiment performed after 2013? Many subtropical experiments have been conducted under conditions that could warrant a comparison, and/or an inclusion.

P2.26 All organisms except phytoplankton? Under which circumstances? What do you mean by "high rate of nutrient and carbon cycle". To what processes are you referring? Please refine and rephrase.

P5.8 Please provide a range of GoM temperature variations.

P5.9 What do you mean by "this is the mean characteristic for using an auxiliary heating or cooling system"? Also, please fix the references.

P5.18 Confusing sentence.

P5.21 Confusing sentence.

P5. How were the mesocosms immersed in water exactly? It is stated that some tests were performed with both treatments (immerged + thrusters),but the text doesn't agree with figure 6 caption. (Fig 6c or 6d is immerged?)

P6.1 "while immersing in water the mesocosms" should be "while immersing the mesocosms in water"

P6.27 The accumulation of sediments is not necessarily a disadvantage. It depends on what processes or parameters you intend to measure. For example, the resuspension of normally sedimenting POM can increase the rates of attached bacterial degradation in the water column which would not have happened normally. Thus, if this is what you intend to measure, an artificially mixed tank will bias your observations, although I agree that an homogeneous tank can be appealing under other circumstances.

Section 2.3.1 Is the propeller upwelling or downwelling water through the center of the

tank? How could that difference affect mixing and or sedimentation? (Slower center-upwelling thruster could allow some sedimentation depending of the mesocosm shape because of the downwelling on the edges.)

P7.22 & P7.28 Repetition.

On many occasions, in-text citations are not properly reported or oddly inserted (too many parentheses, misplaced commas, etc). Please fix.

In-text figure citation should of figures and figures labels could be improved.

P2.18. The sentence has no parentheses at the end. (i.e That make . . .

P2.22 Temperature control is probably the primary challenge with on-land mesocosm experiments due to the thermal inertia of the aquatic environment (Leblud et al., 2014).

P4.23 Odd sentence.

Fig6 caption: Please consider plotting all the graphs using the same y-axis. Also revise caption and y-axis label.

P5.33 "inmersing" should be immersing.

P6.2 Figure 6 e) appears twice

P6.6. What do you define as the "instantaneous variation of maximum temperature"?

P6.6-11 What is the mean temperature change under each scenario? Is stratification affected by the mesocosm immersion ?

P6.14 The appropriate speed to eliminate the stratification could be saved for the next section (on thruster velocity).

---

## Author Comment (AC2) · 23 May 2019

22 may 2019

Prof. Jean Pierre Gattuso
Associated Editor
Biogeosciences

Dear professor Jean Pierre Gattuso

You will find herewith-enclosed the revised version of the manuscript entitled "Technical note: low cost mesocosms design for suties of tropical marine environments", by Rube R. Raygosa-Barahona, Sebastien Putzeys, Jorge Herrera-Silveira and Daniel Pech. The comments of the reviewer have been counterchecked and considered to improve the manuscript. You will find attached our reply to the points raised by the reviewer and how we have dealt whit each one of them. We thank you in advance for your assistance.

Sincerely,

Ruben Raygosa-Barahona

**Manuscrit ID: bg-2019-56**

**Low Cost Mesocosms Design for Studies of Tropical Marine Environments**

Ref P1. Line 7: Mesocosms are often deployed in situ, thus are not necessarily an alternative to in situ studies.

Response: The first sentences in the summary were re written in order to avoid confusion.

Changes: *The design of mesocosms, to conduct enclosed marine experimental studies, imposes diverse challenges associated to technical facilities for maintaining the experimental condition close to natural variability. Here we present and discuss the technical improvements for a mesocosms used to study the effect of oil in the productivity of the tropical marine waters from the Gulf of Mexico*

Ref P2.L4: Is there really no tropical mesocosm experiment performed after 2013? Many subtropical experiments have been conducted under conditions that could warrant a comparison, and/or an inclusion.

Response: The last sentence of the paragraph was modified to highlight challenges imposed when using on lad mesocosmos in tropical environments, also we add some recently published articles . The information of table one was used now to show how the potential effect of the temperature of the enclosed water has been considered.

Changes:  *Most of the on land based mesocosm facilities, located in the northern hemisphere (e.g. MESOAQUA and AQUACOSM websites) has used large artificial enclosures that includes state of the art technologies in order to obtain reliable results.  The technological needs for the implementation of mesocosmos facilities for the study of tropical marine environment poses different challenge du the harsh on land environmental condition associated to high temperature and high evaporation rates. However, the control of the temperature of the enclosed water and its potential effects to bias the results has barely been considered* (Carneiro et al., 2014)(Leblanc et al., 2016)  (Algueró-muñiz, 2017) (Su et al., 2018)(dos Santos Severiano et al., 2018).

*.*

Ref P2.26 All organisms except phytoplankton? Under which circumstances? What do you mean by "high rate of nutrient and carbon cycle". To what processes are you referring? Please refine and rephrase.

Response: The last sentence of the paragraph was rephrased to include more detailed information about the potential effect of the temperature on phytoplankton productivity and its relationship with nutrients and carbon consumption

Changes: *The organisms that typically exhibit higher respiration rates also exhibit higher metabolism rates (Brown et al., 2004). The increment of temperature could induces an increment on the productivity of phytoplankton specific-species by accelerating the nutrient recycling in the water column (Marañon et al. 2018) but also could accelerate the phytoplankton consumption of organic carbon that might reduce the transfer of primary produced organic matter to higher tropic levels (Basu and Mackey 2018).*

Ref P5.8 Please provide a range of GoM temperature variations.

Response: The annual range of temperature variation of GoM was provided

Changes: *The surface temperature (0 – 10 m depth) of the seawater in the south of Gulf of Mexico (GoM)  varies from 22 (november –february) to 28 °C (july-august), but  the ambient temperature could vary from 23°C during the winter frontal storm period (november-february) to 40°C or higher during the dry season (march-may) (Angeles-Gonzales et al. 2017)*

P5.9 What do you mean by "this is the mean characteristic for using an auxiliary heating or cooling system"? Also, please fix the references.

Response: The sentence was modified to put in context the use of a cooling or heating system. The modification is now linked to the preceding sentence

Changes: *This particularity can drastically modify the seawater temperature on the mesocosmos container. To avoid this effect, the use of an auxiliary heating or cooling system is suggested.*

Ref P5.18 Confusing sentence.

Response: The sentences was modified to avoid confusion.

 Changes: *Temperature tests.*

Ref P5.21 Confusing sentence.

Response: The sentences was modified to avoid confusion.

Changes:

*2.2 Temperature Tests.*

*To evaluate the variability of the temperature of the water column on the mesocosm container, a series of measurement were done with and without the effect of the electric thruster and with the mesocosmos container immersed in a water tank and out of it.*

Ref P5. How were the mesocosms immersed in water exactly? It is stated that some tests were performed with both treatments (immerged + thrusters),but the text doesn't agree with figure 6 caption. (Fig 6c or 6d is immerged?)

Response: Both the paragraph and the figure caption were rewritten clarify the text

Changes*:. At time t = 12:24:30 the electric thruster is turned on at 20% of their maximum power. It was observed that the system takes approximately 1 minute to homogenize, even though the external temperature continues to increase.*

*2.2.2 Immersing the mesocosms in a tank water to increase their thermal capacity.*

*To evaluate the effect of immersing the mesocosms inside a water tank different tests were performed:*

- *the mesocosms were exposed to the environment to observe the temperature variations as well as the stratification.*
- *the mesocosms were immersed in water tank.*
- *the effects of the thruster were included, while immersing the mesocosms in a water tank.*
- *the thruster was used with different on / off cycles in.*

*As a first step the on land mesocosmos was left to evolve following the climatic conditions, Figure 6 b) shows this evolution during the day-night temperature cycle and the development of a stratification throughout the day period*

In caption: *d) effect of the thruster functioning continuously at 20% of maximum power, the stratification practically vanished.*

P6.1 "while immersing in water the mesocosms" should be "while immersing the mesocosms in water"

Response: the sentence was corrected

- Changes: *the effects of the thruster were included, while immersing the mesocosms in a water tank.*

P6.27 The accumulation of sediments is not necessarily a disadvantage. It depends on what processes or parameters you intend to measure. For example, the resuspension of normally sedimenting POM can increase the rates of attached bacterial degradation in the water column which would not have happened normally. Thus, if this is what you intend to measure, an artificially mixed tank will bias your observations, although I agree that an homogeneous tank can be appealing under other circumstances.

Response : The accumulation of sediment in our specific case is a severe disadvantage as the future goal is the simulation of a close nearshore environment (less than 2m depth). The use of the mixing system at low speed simulate the natural mixing and maintain a homogeneous water mass as it could be observed in natural conditions. Also, the beneficial effect of the mixing reduced the exposure time of the cells to high UVBR irradiances that could be measured at our latitudes.

Changes: *The use of marine thrusters has the additional advantage of avoiding the accumulation of sediments, which in our specific case (to test the oil exposure effect in marine productivity) is a severe disadvantage as the future goal is the simulation of a close nearshore environment (less than 2m depth). The mixing system at low speed could be used to simulate the natural mixing and maintain a homogeneous water mass as it could be observed in natural conditions.*

Section 2.3.1 Is the propeller upwelling or downwelling water through the center of the C2 BGD Interactive comment Printer-friendly version Discussion paper tank? How could that difference affect mixing and or sedimentation? (Slower centerupwelling thruster could allow some sedimentation depending of the mesocosm shape because of the downwelling on the edges.)

Response :  we agree with the referee.  Our mesoscoms design could be easily configured for centerupwelling or centerdownwelling. Anyway all of the test presented were effectuated using the downwelling configuration because for the oil exposure experimentation the scientific staff considered that avoiding sedimentation would be appropriate for their particular experiment.

Changes:  we add the information on discussion section *:*
*The direction of the propeller rotation could be easily configured for upwelling or downwelling effects by only reversing the electrical connections of the thruster. However all   tests presented were effectuated using the downwelling configuration.  We consider that with a high enough velocity the thruster configuration for upwelling could avoid sedimentation while at low velocities a centerdownwelling thruster could allow sedimentation.*

 P7.22 & P7.28 Repetition. On many occasions, in-text citations are not properly reported or oddly inserted (too many parentheses, misplaced commas, etc). Please fix. In-text figure citation should of figures and figures labels could be improved.

Response :  Comas, parenthesis, speaces, citation where fixed all along the text.  Citation wer inserted using the plugging provided by the journal.

Changes:  we replace the former P7.28 sentence with:

*In addition, CFDs can be used to estimate the profile of velocities into the mesocosms, useful to predict the stresses that the use of propellers can produce to the communities under study.*

P2.18. The sentence has no parentheses at the end. (i.e That make . . .

Response: the paragraph was rephrased and parentheses were eliminated.

Changes: *That make it imperative to take into account some specific considerations prior to the experimental phase*

P2.22 Temperature control is probably the primary challenge with on-land mesocosm experiments due to the thermal inertia of the aquatic environment (Leblud et al., 2014).

Response:  The sentence was modified according to referee coment

Changes:  *Temperature control is probably the primary challenge with on-land mesocosm experiments due to the thermal inertia of the water (Leblud et al., 2014).*

P4.23 Odd sentence

Response:  Sentence was re written in order to avoid confusion

Changes:  *Due to the importance of temperature in altering the rates of metabolism, it is necessary that the temperature in the mesocosms be maintained within the range in which it would be in the ecosystem to be studied*

Fig6 caption: Please consider plotting all the graphs using the same y-axis. Also revise caption and y-axis label.

Response : All graphs were re-drawn using now the same y-axis. Also the y-axis label was corrected

Changes:

[Figure]

P5.33 "inmersing" should be immersing.

Response:  the word was correted.

Change:  *The word "immersing" was corrected in the entire text*

P6.2 Figure 6 e) appears twice

Response:  The paragraph was rewritten and duplicity of "Figure 6e" was eliminates

Change:  *Figure 6e shows the temperature behavior when using  propellers with 5 minute on/off cycle. It can be noted that superficial layer started to heat during this short time, compared to the deeper layer. Figure 6f shows the experiment with the motor at 20 % of its maximum power with a duty cycle of 10 minutes*

P6.6. What do you define as the "instantaneous variation of maximum temperature"?

Response:  we refers to a rapid increment of temperature. The sentence was re written to avoid confussion

Change: *It can be noted that the stratification had a duration of 10 hours with a rapid increase of temperature of about 3°C and a maximum change interval of 4 °C.*

P6.6-11 What is the mean temperature change under each scenario? Is stratification affected by the mesocosm immersion ?

Response:  The temperature in each scenario showed a variation about 15 *°C (maximum value) on land mesocosm without a thruster and a variation about 4°C* on the mesocosm immersed on a water tank with a thruster.

Change:  Information of temperature variability was included in the discussion

: *Immersing the mesocosms in water will reduce the range of  variation of the water temperature compared with the case in which the mesocosms was leaved out of the water, while has no practical effect in the stratification.  The Thruster inclusion only has effect in avoiding the stratification  efficiently on a tropical offshore mesocosms system while do not have a significant effect on the temperature*

P6.14 The appropriate speed to eliminate the stratification could be saved for the next section (on thruster velocity).

Response : *We have decided to leave that statement in its original place to match with the chronological order of the experiments*

**Technical Note: Low Cost Mesocosms Design for Studies of Tropical Marine Environments**

Ruben R. Raygosa-Barahona[1], Sebastien Putzeys[1], Jorge Herrera-Silveira[1], Daniel Pech[2]

[1]Cinvestav, Merida,97600,Mexico

[2]Laboratorio de Biodiversidad Marina y Cambio Climático, Ecosur,Campeche,24500, Mexico

*Correspondence to*: Ruben R. Raygosa-Barahona (r.raygosa@gmail.com)

**Abstract.** The design of mesocosms, to conduct enclosed marine experimental studies, imposes diverse challenges associated to technical facilities for maintaining the experimental condition close to natural variability. Here we present and discuss the technical improvements for a mesocosms used to study the effect of oil in the productivity of the tropical marine waters from the Gulf of Mexico. The mesocosm was equipped with an electric marine thruster as a means of avoiding stratification in the water contained in it. In addition, the system was submerged in a water tank to increase the thermal inertia and maintain the temperature variations within reasonable ranges. The design does not include auxiliary forcing cooling systems. The results revealed the influence of climatology on the mesocosms' temperature and showed the feasibility of the proposed design for tropical environments. With high variations of ambient temperature (>20ºC, during the day), the variations in the mesocosm temperature were only 3ºC. The range of temperature variations were similar to those that occur in tropical environments from the Gulf of Mexico.

**1 Introduction**

*In situ* ocean environmental studies are expensive and often difficult to perform (Reilly, 1999). On the other hand, the results of small-scale laboratory experiments are cheap, but they do not incorporate sufficient environmental realism (Cappello and Yakimov, 2010). The mesocosms studies are an intermediate alternative between both laboratories and *in situ* experiments. The ability to control environmental variables (Vallino, 2000) jointly with the replicability and repeatability compared to field studies (Cappello and Yakimov, 2010; Petersen and Hastings, 2001)(Auffan et al., 2018) convert the mesocosm into a powerful experimental option. Best of all, parts (populations) and wholes (ecosystems) can be investigated simultaneously (Odum, 1984). The terms micro-, meso- and macrocosms are often used to define multifactorial ecosystems studies. A volume classification has been suggested in which microcosms are enclosures smaller than 1 m$^3$, mesocosms to contain between 1 to 1000 m$^3$, and finally macrocosms to contain more than 1000 m$^3$ (Culp et al., 1988). The material and design used in these cosms constructions usually depend on the experiment's localization and the objective of the study. Enclosures with flexible walls are commonly used *in situ*, whereas containers with rigid walls are used either on shore or in indoor facilities. Mesocosms experimental approach is arguably the single most powerful method to obtain a mechanistic quantitative understanding of

ecosystem-level impacts of stressors on complex systems. Most of the on land based mesocosm facilities, located in the northern hemisphere (e.g. MESOAQUA and AQUACOSM websites) has used large artificial enclosures that includes state of the art technologies in order to obtain reliable results. The technological needs for the implementation of mesocosms facilities for the study of tropical marine environment poses different challenge due the harsh on land environmental condition associated

5    to high temperature and high evaporation rates. However, the control of the temperature of the enclosed water and its potential effects to bias the results has barely been considered, (Carneiro et al., 2014, Leblanc et al., 2016, Algueró-muñiz, 2017, Su et al., 2018).

In the existing literature, temperature control or register was not considered a priority despite the variation of temperature in tropical areas. The differences in temperature recorded in different studies are highly variable (Table 1), probably causing the

10    presence of a thermocline during the experiment. None of these studies has considered temperature as a relevant factor, probably due to the small impact expected on the objective of the study.

Here we designed and tested a ground mesocosms unit system developed for studies in a tropical environment where the water column temperature profile could be a relevant factor modifying the response of the productivity and metabolic activity of the micro-organisms on the water column. Additionally the atmospheric temperature could induce marked changes in the water

15    column temperature and high rates of evaporation.

The design of a mesocosms experiment is a complex procedure in which the variables and parameters (chemical, physical or biological) have to be balanced with the objectives of the study and the development feasibility (Riebesell et al., 2013). In the case of experiments in tropical areas such as the south of the Gulf of Mexico, the greatest challenge could be the harsh environmental conditions: the high environmental temperatures (32-40 ℃), the high evaporation rates, recurrence of storms

20    and hurricanes (June to November). That make it imperative to take into account some specific considerations prior to the experimental phase. Here we describe and discuss the series of analyses and tests performed to determine the feasibility of mesocosms experiments in the tropical region of the Yucatan peninsula, México.

Metabolism is the process by which energy and mass are transformed within an organism and exchanged between the organism and its environment. Metabolic theory predicts how metabolic rate controls ecological processes (Gillooly et al.,

25    2001) . On the other hand, temperature is a master variable that controls biological activity through its fundamental effect on metabolic rate (Gillooly et al., 1994). Temperature changes affect several physical and chemical properties of water such as conductivity, salinity, density (viscosity), oxygen solubility, pH and carbon dioxide solubility. Also, all the biological processes are temperature dependent. The organisms that typically exhibit higher respiration rates also exhibit higher metabolism rates (Brown et al., 2004). The increment of temperature could induces an increment on the productivity of

30    phytoplankton specific-species by accelerating the nutrient recycling in the water column (Marañon et al. 2018) but also could accelerate the phytoplankton consumption of organic carbon that might reduce the transfer of primary produced organic matter to higher tropic levels (Basu and Mackey, 2018).

[revised manuscript text omitted]

The surface temperature (0 – 10 m depth) of the seawater in the south of Gulf of Mexico (GoM) varies from 22 (november –

15 february) to 28 °C (july-august), but the ambient temperature could vary from 23°C during the winter frontal storm period (november-february) to 40°C or higher during the dry season (march-may) (Ángeles-gonzález, 2017). This particularity can drastically modify the seawater temperature on the mesocosms container. To avoid this effect, the use of an auxiliary heating or cooling system is suggested. The containers used for the ground mesocosms prototype are commercial water tanks made of high-density polyethylene. The container shape is a cylinder 1.75 m high by 1.55 m in diameter, see Figure 5. A commercial

20 brushed NEREAUS marine electric thruster of ½ hp composes the recirculating system. The thruster's velocity is driven by a PWM motor controller controlled either by a microcontroller or manually by a potentiometer adjustable by the user. Note in Figure 5 the temperature column sensor located on the left side. Additionally, the system has a radio link to be able to monitor the data in real time.

**2.2 Temperature Tests.**

25 To evaluate the variability of the temperature of the water column on the mesocosm container, a series of measurement were done with and without the effect of the electric thruster and with the mesocosmos container immersed in a tank of water and out of it.

**2.2.1 Temperature Profiler Tests.**

30 To evaluate the performance of the electric thruster to avoid water stratification in the mesocosmos container, an artificial temperature gradient using two heaters of 2000 W each was created. The heaters were placed 30 cm below the surface for one hour, and 15 temperature sensors were installed vertically aligned along the water column. One of the sensors was installed just above the water level and the second on was installed 30 cm above to measure the ambient temperature. The electric marine thruster was then power on at 25% of its maximum power. Figure 6a) shows the temporal evolution of the temperature

in the container throughout the experiment. From now on for the following figures, S1 to S15 labels and their respective color stand for each one of the signal fetched for 15 temperature sensor installed into the container. The sensors 1 to 13 are submerged in the water in an order according to their number from the deepest to the surface. The sensors 14 and 15, in blue and brow respectively are above the water level. At time t = 12:24:30 the electric thruster is turned on at 20% of their maximum power. It was observed that the system takes approximately 1 minute to homogenize, even though the external temperature continues to increase.

2.2.2 Immersing the mesocosms in a water tank to increase their thermal capacity.

To evaluate the effect of immersing the mesocosms inside a water tank different tests were performed:

- the mesocosms were exposed to the environment to observe the temperature variations as well as the stratification.
- the mesocosms were immersed in water tank.
- the effects of the thruster were included, while immersing the mesocosms in a water tank.
- the thruster was used with different on / off cycles in.

[revised manuscript text omitted]

**3 Discussions and Conclusions**

No literature describing the use of electric propellers in mesocosms was found for tropical on land mesocosms. Without the use of a thruster, a significant temperature difference appears between superficial and bottom layers during a 24h period. This temperature difference could promote micro-conditions (Nada Krstulovic, at al 1995) that would alter biological communities' responses during an experiment, mainly when performed on bacterioplankton and phytoplankton (Lorena Grubisic at al 2012). Immersing the mesocosms in a water tank will reduce the range of variation of the water temperature compared with the case in which the mesocosms was leaved out of the water, while has no practical effect in the stratification.  The inclusion of the thruster only has effect to avoiding stratification efficiently on a tropical on land mesocosms system while do not have a significant effect on the temperature. Additionally, its inclusion in the mesocosms design can be useful to reduce the stress of marine communities under study compared to using water recirculation systems.

The direction of the propeller rotation could be easily configured for upwelling or downwelling effects by only reversing the electrical connections of the thruster. However all tests presented here were effectuated using the downwelling configuration. We consider that with a high enough velocity the thruster configuration for upwelling could avoid sedimentation while at low velocities a centerdownwelling thruster could allow sedimentation.

5   The different tests carried out compared to the *in situ* data suggest the variability inside the mesocosms container could be considered as part of the natural daily changes.

In addition, CFDs can be used to estimate the profile of velocities into the mesocosms, useful to predict the stresses that the use of propellers can produce to the communities under study.

The different tests have been carried out under no controlled environmental conditions, Surprisingly, environmental variations
10  of up to 25 °C of temperature resulted in only 3 °**C** of temperature variation in the mesocosms. The authors consider that this variability compared to the *in situ* data suggest the variability inside the tank could be considered as part of the natural daily changes in the environment.

The immersion of the mesocosm unit in the water tank together with the use of a marine thruster shows its efficiency to maintain water temperature relatively stable without the need to add auxiliary refrigeration equipment. This is a lower cost
15  strategy. During the preparation of the different tests, a decrease in water level due to evaporation was observed, averaging 6 mm per day. This reduction in level represents approximately 6 liters of water per day. This variation can produce important changes in some physical parameters such as salinity. Future work may be to implement a system for measuring evaporation. The series of tests performed allow us to conclude that in the case of onshore mesocosms experiments in tropical areas, auxiliary and higly costly refrigeration equipment is not necessary. This allows both lower installation and running costs.

20  **Acknowledgements**

Thanks to our colleagues from the ISMER-UQAR, specially to Gustavo Ferreyra for share it experiences with the implementation of on land mesocosms. And to Ismael Mariñp Tapia for share records from sea temperatures in Telchac Beach. This work was part of the binational collaboration México-Quebec project "Desarrollo de experimentos en mesocosmos para evaluar la vulnerabilidad de los ecosistema marios ocasionada por la actividad petrolera: comparición latitudinal" FONCICyT
25  265435. Financial support was also provided by the SENER2012-1-Hidrocarburos (Sectorial Research Funds 0020SRE-CONACYT S0018) from the Mexican Council of Science and Technology (ConaCyt), the Mexican Secretary of Energy (SENER) and the Mexican Petroleum Company (PEMEX).

30

Algueró-muñiz, M.: Ocean acidification effects on mesozooplankton community development : Results from a long-term mesocosm experiment, , 12(4), 1–29, 2017.

Ángeles-gonzález, L. E.: Molluscan Studies, J. Molluscan Stud., 83(September 2018), 280–288, doi:10.1093/mollus/eyx013, 2017.

[revised manuscript text omitted]